# Performance of the Global Forecast System's Medium-Range Precipitation Forecasts in the Niger River Basin Using Multiple Satellite-Based Products

Haowen Yue[1], Mekonnen Gebremichael[1], and Vahid Nourani[2,3]

[1] Department of Civil and Environmental Engineering, University of California, Los Angeles, CA 90095-1593
[2] Faculty of Civil Engineering, University of Tabriz, Tabriz, Iran
[3] Near East University, Faculty of Civil and Environmental Engineering, Near East Boulevard, 99138, Nicosia, via Mersin 10, Turkey

*Correspondence to*: Mekonnen Gebremichael (mekonnen@seas.ucla.edu)

**Abstract.** Accurate weather forecast information has the potential to improve water resources management, energy,
and agriculture. This study evaluates the accuracy of medium-range (1 – 15 day) precipitation forecasts from the
Global Forecast System (GFS) over watersheds of eight major dams (Selingue Dam, Markala Dam, Goronyo Dam,
Bakolori Dam, Kainji Dam, Jebba Dam, Dadin Kowa Dam, and Lagdo Dam) in the Niger river basin using NASA's
Integrated Multi-satellitE Retrievals (IMERG) "Final Run" satellite-gauge merged rainfall observations. The results
indicate that the accuracy of GFS forecast varies depending on climatic regime, lead time, accumulation timescale,
and spatial scale. The GFS forecast has large overestimation bias in the Guinea region of the basin (wet climatic
regime), moderate overestimation bias in the Savannah region (moderately wet climatic regime), but has no bias in
the Sahel region (dry climate). Averaging the forecasts at coarser spatial scales leads to increased forecast accuracy.
For daily rainfall forecasts, the performance of GFS is very low for almost all watersheds except for Markala and
Kainji dams, both of which have much larger watershed areas compared to the other watersheds. Averaging the
forecasts at longer time scales also leads to increased forecast accuracy. The GFS forecasts, at 15-day accumulation
timescale, have better performance, but tend to overestimate high rain rates. Additionally, the performance assessment
of two other satellite products was conducted using IMERG Final estimates as reference. The Climate Hazards Group
InfraRed Precipitation with Station data (CHIRPS) satellite-gauge merged product has similar rainfall characteristics
with IMERG Final, indicating the robustness of IMERG Final. The IMERG "Early Run" satellite-only rainfall product
is biased in the dry Sahel region, however in the wet Guinea and Savannah regions, IMERG "Early Run" outperforms
GFS in terms of bias.


## 1. Introduction

Global climate forecasts, with lead times ranging from hours to several months, are becoming increasingly available (Saha et al. 2014; Abdalla et al. 2013; NCEP 2015; JMA 2019). Significant societal benefit could be realized from research to reduce common barriers in climate forecast utilization blocking the path to improving water resources management, energy, and agriculture. One such a barrier is the lack of understanding of climate forecast accuracy in different regions of the world. This focus is timely given the recent advances in numerical atmospheric models, and in the wealth of new observing capabilities including satellite remote sensing. These combined models and observational datasets provide opportunity for researchers to quantify the accuracy of climate forecasts.

The Niger River is the principal river of West Africa, and is shared among nine riparian countries (Fig. 1): Benin, Burkina Faso, Cameroon, Chad, Guinea, Ivory Coast, Mali, Niger and Nigeria. The basin is facing multiple pressures from increasing population, water abstraction for irrigation, and risk of extreme hydrological events due to climate change (Sylla et al. 2018). A number of hydropower dams exist in the region, and additional dam projects are envisaged in order to alleviate chronic power shortages in the countries of the Niger basin. Optimal management of water resources is key to maximizing benefits, such as hydropower generation, and minimize disasters, such as flooding. Climate forecast information has the potential to improve water resources management, energy, and agriculture (e.g., Patt et al. 2007; Breuer et al. 2010; Mase and Prokopy 2014; Pandya et al. 2015; Koppa et al. 2019; Alexander et al. 2020). For example, in a recent study, Koppa et al (2019) showed that the use of seasonal precipitation forecasts in reservoir planning of Omo Gibe dam in Ethiopia can increase annual hydropower generation by around 40%.

Several studies have investigated the accuracy of seasonal forecasts in West Africa (e.g., Bliefernicht et al. 2019; Pirret et al. 2020). Seasonal forecasts are important for water resource planning, while medium-range (1-day to 15-day) forecasts are important for operational decisions, such as reservoir operations. The availability of medium-range global climate forecasts has grown in recent years. Examples of such forecast products include Global Forecast System (GFS; NCEP 2015), NCEP climate forecast system (NSF CFS, Saha et al. 2014), European Centre for Medium-Range Weather Forecasts (ECMWF; Abdalla et al. 2013), and Global Spectral Model (GSM; JMA 2019). For these precipitation forecasts to be effectively used in applications, their accuracy must be known, which is usually performed

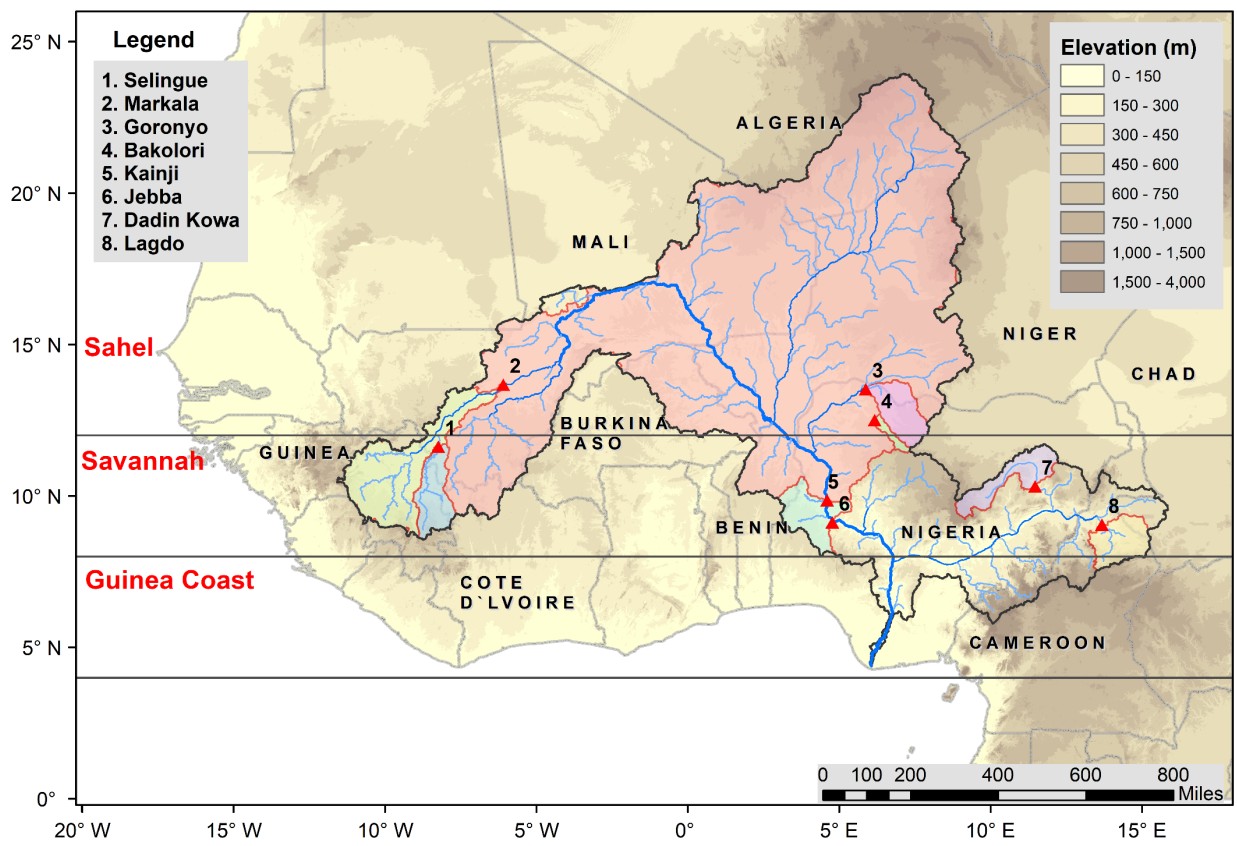

Figure 1. The Niger River Basin, and locations of major reservoir dams in the basin: (1) Selingue, (2) Markala, (3) Goronyo, (4) Bakolori, (5) Kainji, (6) Jebba, (7) Dadin Kowa, and (8) Lagdo.

through comparison of precipitation forecasts to observations (e.g., Tian et al. 2017; Yuan et al. 2014). Wang et al.
(2019) performed numerical experiment to examine the sensitivity of GFS to inclusion or exclusion of additional
observations collected over the eastern Pacific during the El Niño Rapid Response (ENRR) field campaign, type of
data assimilation method to prepare the initial conditions, and inclusion or exclusion of stochastic parameterizations
in the forecast model. They reported that the GFS forecast errors are only slightly sensitive to the additional ENRR
observations, more sensitive to the DA methods, and most sensitive to the inclusion of stochastic parameterizations in
the model. In addition, they reported that GFS forecasts have difficulty to capture the location and magnitude of heavy
rain rates. Sridevi et al. (2018) evaluated the performance of GFS in India by using rain gauge and satellite rainfall
product, and reported that the GFS forecast shows some skills in 1-day and 2-day lead times, but low skills from 3-
day onwards. Lien et al. (2016) compared the statistical properties of GFS forecasts and Tropical Rainfall Measuring
Mission (TRMM) Multisatellite Precipitation Analysis (TMPA; Huffman et al. 2007, 2010) observations.  They
reported that the GFS model has positive bias in precipitation amount compared to TMPA observations, and that the
GFS forecasts have large random errors at higher resolutions, especially for convective precipitation. According to
Jiang et al. (2015) the lack of consideration of the Aerosol-Cloud Interactions (ACIs) in the GFS model leads to
significant bias in the GFS precipitation forecasts.

In our study region of the Niger River basin, there has not been any performance evaluation of GFS precipitation
forecasts to date. The Niger basin lies in three different climate regimes (wet regime, moderately wet regime, and dry
regime), and is home to nine major irrigation and hydropower dams (Selingue, Markala, Goronyo, Bakolori, Kainji,
Jebba, Dadin Kowa, and Lagdo). Recent advances in satellite rainfall products, particularly following the Global
Satellite Measurement satellite mission (GPM; Hou et al. 2014), and extensive evaluation of GPM rainfall products
in West Africa, provides us with opportunity to use GPM rainfall products as reference for evaluation. Many studies
have conducted to evaluate the accuracy of the satellite rainfall estimates in West Africa. Dezfuli et al. (2017a)
evaluated the performance of NASA's Integrated Multi-satellitE Retrievals (IMERG) "Final Run" (IMERG Final)
(version 4; Huffman et al. 2019a, b) in comparison with two, high-resolution, experimental rain gauge station data
provided by the Trans-African Hydro-Meteorological Observatory (TAHMO; van de Giesen et al. 2014), and reported
the capability of IMERG Final to represent well the diurnal cycle of rainfall. Using the same dataset, Dezfuli et al.
(2017b) showed that IMERG Final is able to capture the propagation of large Mesoscale Convective Systems (MCSs),
a significant advantage over its predecessor's (TMPA) 3-hourly temporal resolution, which misses the time evolution
of most of these systems. Gossett et al. (2018) evaluated the performance of a number of satellite rainfall products
(focusing only on versions that do not include rain gauge data) by comparison with rain gauge station networks in
Benin and Niger, and reported that the satellite products (especially IMERG Early) exhibit high performance in Niger
but relatively lower performance in Benin. Satgé et al. (2020) evaluated the accuracy of a number of gridded
precipitation datasets over West Africa through comparison against rain gauge station data, and reported that CHIRPS
and TMPA (the predecessor to IMERG) provided reliable estimates at both daily and monthly timescales, while the
remaining satellite products considered (CMORPH, PERSIANN, GSMaP, ARC, and TAMSAT) and all atmospheric
reanalysis products considered (MERRA and JRA) were deemed unreliable. Furthermore, they found out that satellite
products that incorporated rain gauge information outperformed satellite-only products. Maranan et al. (2020)
compared IMERG Final products against experimental rain gauge station data in the moist forest region of Ghana,
West Africa, and showed that IMERG Final datasets are able to capture monthly rainfall with a correlation coefficient
close to unity.

The objective of this study is to evaluate the accuracy of medium-range precipitation forecasts derived from the Global
Forecast System (GFS) for the major reservoir dams of the Niger basin through comparison against IMERG Final.
We chose GFS model due to its relatively high spatial (0.25° x 0.25°) and temporal resolution (3-hourly to 6-hourly)
as well as free-of-charge data availability to users. The main questions addressed in this study are as follows. First,
how does the accuracy of GFS forecast vary across different reservoir dams in the same basin? Second, how does the
accuracy vary with lead time in the range 1- to 15-day? Third, what is the effect of spatial averaging (from 0.25° all
the way to basin-scale) and temporal aggregation (from 1-day to 15-day) on the forecast accuracy? Fourth, how does
the accuracy of GFS forecast compare with the accuracy of satellite-only rainfall products that are available in near-
real-time, as the latter may have the potential to calibrate and improve the accuracy of GFS?

**2.   Data and Methodology**
**2.1 Global Forecast System (GFS) Medium-Range Precipitation Forecasts**

The Global Forecast System (GFS) is a global numerical weather prediction system run by the U.S. National Weather
Service (NWS). The GFS forecast products with a resolution of 0.25° by 0.25° are obtained from National Center for
Atmospheric Research (NCAR) Research Data Archive (RDA) GFS Historical Archive (NCEP 2015). The GFS is
run four times a day at UTC 00, UTC 06, UTC 12, and UTC 18 hours. One of the GFS model output variables is
accumulated precipitation, where the precipitation forecasts are accumulations starting from the model run time. We
obtained the 1-day lead daily rainfall forecast by subtracting the 24-hour rainfall accumulation forecast from the 48-
hour rainfall accumulation forecast. Similarly, in order to obtain the 5-day lead daily rainfall forecast, we subtracted
the 120-hour rainfall accumulation forecast from the 144-hour rainfall forecast. We only considered the model runs at
UTC 00 hour.

The GFS model went through a major upgrade, and its version-15 forecasts are available since June 12, 2019. In
version 15, the Finite Volume Cubed Sphere dynamical model (FV3) replaced the Global Spectral Model (GSM) as
the core model. In the GSM model, the horizontal resolutions were T1543 (12.5km) from 0 to 240 hours (0-10 days)
and T574 (~34km) from 240 to 384 hours (10-16 days) (NCEP 2021a). However, in the FV3 model, the horizontal
resolution of the model is about 13 km for days 0-16 (NCEP 2021b). The model runs are re-gridded to produce
precipitation forecasts at 0.25° resolution (NCEP 2015). The Key FV3 model schemes include (Putman and Lin 2007):
(1) the Rapid Radiative Transfer Method for GCMs (RRTMG) scheme for shortwave/longwave radiation (Mlawer et
al. 1997; Iacono et al. 2000; Clough et al. 2005), (2) the Hybrid eddy-diffusivity mass-flux (EDMF) scheme for
Planetary Boundary Layer (PBL) (NCEP, 2021a), (3) the Noah Land Surface Model (LSM) scheme for land surface
option (Chen et al. 1997), (4) the Simplified Arakawa-Schubert (SAS) deep convection for cumulus parameterization
(Arakawa et al. 1974; Grell 1993), and (5) an advanced GFDL microphysics scheme for microphysics (NCEP, 2021b).

**2.2 IMERG Final Satellite Precipitation Products**
IMERG Final rainfall products are used in this study as reference to evaluate the performance of GFS precipitation
forecasts. IMERG Final combines all available microwave precipitation estimates, microwave-calibrated infrared
estimates, and rain gauge data to provide rainfall estimates at very high resolution (30-minute, 0.10°) (Hou et al. 2014;
Huffman et al. 2015). The IMERG products are categorized into three types, namely early run, late run, and final run.
It is only the final run or "final" version that incorporates rain gauge data. The data latency of IMERG Final is about
3.5 months. Details of IMERG algorithm developed by NASA are available at Huffman et al (2019a, b). The latest
version (V6B) of IMERG datasets have been accessed from the NASA's Earth Data Goddard Earth Sciences Data
and Information Services Center (GES DISC) web portal.

**2.3 Other Satellite Precipitation Products**
In order to put the GFS forecast performance into perspective, we also evaluated two other state-of-the-art satellite
rainfall products:
• IMERG Early provides un-calibrated IMERG rainfall fields, which do not include correction from rain gauges.
The data latency of IMERG Early is near-real-time, about 4 hours. We have used the latest version (V6B) of
IMERG Early datasets. Post-processing calibration of GFS forecasts (in order to improve the accuracy of GFS
forecasts) requires the use of "relatively better performing" and "available in near-real-time" independent
rainfall observations to correct real-time dynamical GFS model forecasts. Comparison of the performance of
IMERG Early with the performance of GFS would indicate to what extent the IMER Early products could be
used for calibration of GFS forecasts.
• The Climate Hazards Group InfraRed Precipitation with Station data (CHIRPS) is derived primarily from
thermal infrared data using the cold cloud duration (CCD) approach, calibrated using TRMM Multi-satellite
Precipitation analysis (TMPA 3B42 v7; Huffman et al. 2007) precipitation datasets by local regression, and
include rain gauge station data from multiple sources (regional and national meteorological services). CHIRPS
data are available at a spatial resolution of 0.05° and a temporal resolution of 1-day, with a data latency period
of about 3 weeks. Details of CHIRPS algorithm are available at Funk et al. (2015).  Agreement between the
reference (IMERG Final) and CHIRPS would indicate that the IMERG Final estimates are robust.

**2.4 Study Region**
The Niger river, with a drainage basin of 2,117,700 $Km^2$, is the third longest river in Africa. The source of the main
river is in the Guinea Highlands, and runs through Mali, Niger, on the border with Benin and then through Nigeria,
discharging through a massive delta, known as the Niger Delta (the world's third largest wetland), into the Atlantic
Ocean. The rainfall regimes in the region follow the seasonal migration of the Inter-Tropical Convergence Zone
(ITCZ), which brings rainfall primarily in the summer season (Animashaun et al. 2020; Sorí et al. 2017).
Climatologically, the Niger basin  lies in three latitudinal sub-regions (Akinsanola et al. 2015, 2017): (1) the Guinea
coast (latitude 4°–8°N), which borders the tropical Atlantic Ocean in the south; (2) the Savannah (latitude 8°–12°N),
an intermediate sub-region; and (3) the Sahel (latitude > 12°N) to the north. The Guinea coast experiences  a
bimodal rainfall regime that is centered in the summer monsoon period of June–September, with August being the
period of a short dry season, while the Savannah and Sahel sub-regions experience a unimodal rainfall regime, with
maximum rainfall occurring in August (Akinsanola and Zhou 2018). The ranges of annual rainfall  amounts are: 400–
600 mm in the Sahel, 900–1200 mm in the Savannah; and 1500–2000 mm in the Guinea coast (Akinsanola et al.

180  2017).


The Niger basin is home to eight major reservoir dams (see Table 1 and Fig. 1): (1) Selingue Dam in Mali: a primarily
hydropower dam, (2) Markala Dam in Mali: a primarily irrigation dam, serving about 75,000 ha of farmland, (3)
Goronyo Dam in Nigeria: a multi-purpose dam for flood control, provision of downstream water supply and the release
of water for irrigation in the dry season, (4) Bakolori Dam in Nigeria: a primarily irrigation dam with a command area
of about 23,000 ha, (5) Kainji Dam in Nigeria: the largest Dam on the Niger supplying power for most towns in
Nigeria, (6) Jebba Dam in Nigeria: a primarily hydropower dam, (7) Dadin Kowa Dam:  a multi-purpose dam for
water supply, electricity and irrigation, (8) Lagdo Dam in Cameroon: a multi-purpose dam providing electricity to the
northern part of the country and supplying irrigation water for 15,000 hectares of cropland. The watersheds of the
dams are primarily either in the Savanna (Selingue, Markala, Jebba, Dadin Kowa, an Lagdo), or in the Sahel (Goronyo,
Kainji), or partly in both (Bakolori). The watershed sizes vary over a large range, from 4,887 $Km^2$ (Bakolori Dam) to
1,464,092 $Km^2$ (Kainji Dam). The average elevations of the watersheds are close to each other at $500 \pm 50$ m.a.s.l.

In order to make the results of this study meaningful to reservoir managers, the Niger basin was divided into
watersheds according to the locations of the dam reservoirs (see Fig. 1). Then the sub-basin of each dam was defined
as the drainage between the dam itself and the upstream dam. For example, the drainage basin of the Markala Dam
does not include the drainage basin of the Selingue Dam.

Table 1. Selected dams and their watershed characteristics

| Dam | Country | Operational since* | Capacity (million m³)* | Power (MW)* | Primary Purpose* | | | Area of Drainage Basin (km²)** | Elevation of Drainage Basin (m)** |
|-----|---------|-------------------|------------------------|-------------|------------------|---|---|-------------------------------|-----------------------------------|
| | | | | | Irrigation and Water Supply | Flood Control | Hydroelectricity | | |
| Selingue | Mali | 1982 | 2170 | 44 | | | x | 32685 | 473 |
| Markala | Mali | 1947 | 175 | | x | | | 102882 | 442 |
| Goronyo | Nigeria | 1983 | 942 | | x | x | | 31547 | 446 |
| Bakolori | Nigeria | 1978 | 450 | | x | | | 4887 | 519 |
| Kainji | Nigeria | 1968 | 15000 | 960 | | | x | 1464092 | 406 |
| Jebba | Nigeria | 1984 | 3600 | 540 | | | x | 40268 | 308 |
| Dadin Kowa | Nigeria | 1988 | 2855 | 35 | x | | x | 32936 | 535 |
| Lagdo | Cameroon | 1983 | 7800 | 72 | | x | x | 31352 | 452 |

* information obtained from the Global Reservoir and Dam Database (Lehner et al. 2011) and Food and Agriculture Organization
of the United Nations (FAO)'s Global Information System on Water and Agriculture (AQUASTAT).
** Calculated from HydroSEHDS (Lehner et al. 2008).
**2.5 Evaluation**
IMERG Final rainfall products are used in this study as reference to evaluate the performance of GFS precipitation
forecasts. The comparison period is 15 June 2019 to 15 June 2020 to match the period for which the version-15 of
GFS model forecasts is available. The spatial resolutions of the forecast and satellite products are different: 0.25°(GFS),
0.10° (IMERG Final and IMERG Early), and 0.05° (CHIRPS). The temporal resolutions of the satellite products are:
30-minute (IMERG Final and IMERG Early) and daily (CHIRPS). Our comparison is mostly based on sub-basin (i.e.
watershed for each dam) average values, in which case we average all the datasets to the sub-basin spatial scale. In
some cases, where we compare the spatial patterns of rainfall, we resample both IMERG products and CHIRPS to
0.25° using the bilinear interpolation technique to match the spatial resolution of GFS.

For evaluation metrics, we used the modified Kling-Gupta Efficiency (KGE; Gupta et al. 2009; Kling et al. 2012) and
its components: Bias Ratio (BR), correlation (R), and variability ratio ($\gamma$). *KGE* measures the goodness-of-fit between
estimates of precipitation forecasts and reference observations as:
$$KGE = 1 - \sqrt{(R-1)^2 + (BR-1)^2 + (\gamma-1)^2},$$
$$BR = \frac{\mu_f}{\mu_o},$$
$$\gamma = \frac{CV_f}{CV_o},$$
where R is the linear correlation coefficient between forecasted and observed precipitation, BR is the bias ratio, $\gamma$ is
the variability ratio, $\mu$ is the mean precipitation, CV is the coefficient of variation, and the indices *f* and *o* represent
forecasted and observed precipitation values, respectively. KGE values range from -∞ to 1, with values closer to 1
indicating better model performance. Towner et al. (2019) suggested the following classifications: "Good" (KGE ≥
0.75), "Intermediate" (0.75 ≥ KGE ≥ 0.5), "Poor" (0.5 ≥ KGE > 0), and "Very poor" (KGE ≤ 0). The *BR* values
greater than 1 indicate a positive bias whereby forecasts overestimate precipitation relative to the observed data,
while values less than 1 represent an underestimation. *The $\gamma$ values* greater than 1 indicate that the variability in the
forecast time series is higher than that observed, and values less than 1 show the opposite effect. The *R* measures the
strength and direction of the linear relationship between the forecast and observed values, and to what extent the
temporal dynamics of observed rainfall is captured in the forecasts. The correlation values of 0.6 or more are
considered to be skillful (e.g., Alfieri et al. 2013). In addition, the root mean-square-error normalized by reference
precipitation mean (NRMSE) was also used.

## 3. Results and Discussion

### 3.1 Annual Spatial Variability and Seasonal Characteristics

The spatial map of annual (15 June 2019 – 15 June 2020) rainfall from the various rainfall products is given in Figure 2. According to the reference rainfall product (i.e. IMERG Final), the Niger basin experiences average annual rainfall of 700 mm. The spatial rainfall distribution shows north-to-south increasing gradient, with the Sahel region (> 12°N) receiving on average 346 mm per year, the Savanna region (8°N – 12°N) receiving on average 1,206 mm per year, and the Guinea region (4°N – 8°N) receiving on average 1,620 mm per year. The spatial structures (climatology and north-south gradient in rainfall) of GFS, IMERG and CHIRPS rainfall fields are quite similar to those of IMERG Final. However, the 1-day GFS tends to overestimate in the wet Guinea region of the basin, whereas both IMERG Early and CHIRPS give values that are very close to IMERG Final.

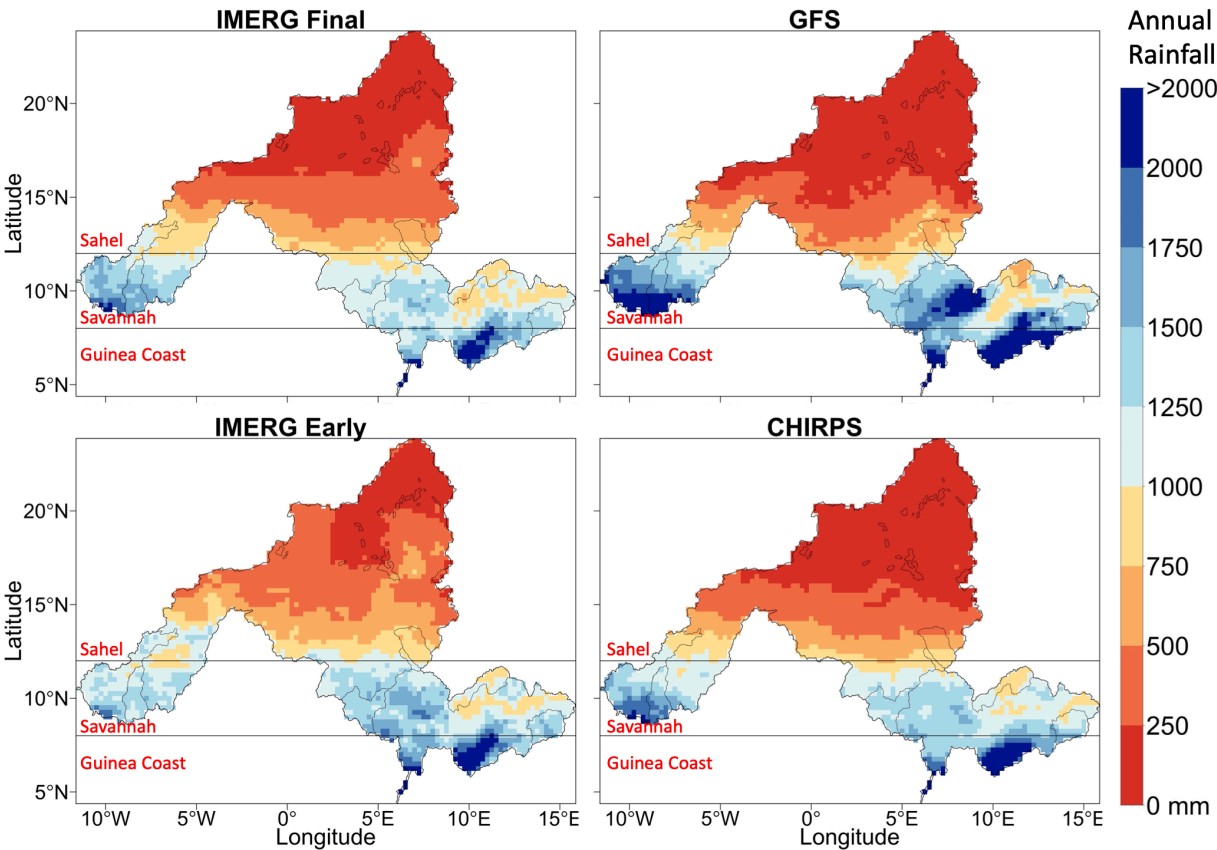

Figure 2. Spatial map of annual rainfall (in mm), for the period 15 June 2019 to 15 June 2020, derived from (a) IMERG Final, (b) GFS (1-day lead time), (c) IMERG Early, and (d) CHIRPS.

Figure 3 shows the seasonal rainfall pattern for each climatological region. According to the reference IMERG Final,
as one goes from north to south, the rainy season expands from 3 months (June – September) in the Sahel to 6 months
(March – November) in the Savanna and Guinea regions. The peak rainfall also shows north-south gradient, with peak
rainfall of 130 mm in the Sahel, to 269 mm in the Savanna, and 350 mm in the Guinea. The rainfall pattern is unimodal
with a peak rainfall value in August for both Sahel and Savanna, but becomes bimodal with one peak in May and the
other in September for Guinea.

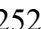

Figure 3. Monthly precipitation regime for the three climatological zones of the Niger river Basin: (a) Sahel, (b) Savanna, and (c) Guinea. Analyses are based on rainfall fields derived from IMERG Final, 1-day-lead GFS. IMERG Early, and CHIRPS. The time period covers from 15 June 2019 to 15 June 2020.


When validated against IMERG Final, the performance of GFS in capturing the seasonal rainfall characteristics
deteriorates as one goes from north to south. GFS captures both the seasonal rainfall pattern and rainfall peak in the
Sahel, and captures the seasonal rainfall pattern but tends to moderately overestimate the peak in the Savannah, while
it has large overestimation (almost twice as much as the reference) in the Guinea particularly during summer. As far
as the other satellite products are concerned, IMERG Early tends to slightly overestimate in the Sahel across all rainy
months, but performs relatively well in the Savannah and Guinea regions. CHIRPS is very close to IMERG Final in
all regions and months, with the exception of modest overestimation of the July rainfall in Guinea.

**3.2 How well do GFS forecasts capture annual rainfall?**
Here, we aggregate the 1-day lead GFS forecasts to annual time scale and compare the results against corresponding
annual precipitation estimates from IMERG Final. Figure 4 presents the watershed-averaged annual rainfall for each
dam watershed. According to IMERG Final, the annual rainfall varies from 434 mm (in Kainji) to 1,481 mm (in
Selingue). Watersheds 1 (Selingue) and 2 (Markala), located in the western part of the Savannah, receive the largest
amount of rainfall, i.e., 1481 mm and 1406 mm, respectively. Watershed 3 (Markala), located in the eastern part of
the Sahel, receives 741 mm of annual rainfall. Watershed 4 (Bakolori), characterized by the smallest watershed area
compared to the rest of the watersheds, lies partly in the Sahel and partly in the Savannah region and receives 921 mm
of annual rainfall. Watershed 5 (Kainji), characterized by the largest watershed area of all, lies mostly in the Sahel
region and receives the lowest amount of annual rainfall (434 mm). Watersheds 6 (Jebba), 7 (Dadin Kowa), and 8
(Lagdo), located in the Savannah, receive annual rainfall amounts of 1190 mm, 941 mm, and 1295 mm, respectively.

Validated against IMERG Final, the GFS tends to overestimate rainfall in all watersheds located in the Savannah (or
watersheds that receive relatively large rainfall amounts), with an overestimation varying in the range 8% to 33%,
with larger bias for watersheds receiving higher rainfall amount. For watersheds in the Sahel (watersheds receiving
low rainfall amount), GFS gives less bias (-11% for the driest Kainji watershed and +10% for Bakolori).

In contrast, IMERG Early tends to underestimate rainfall in all watersheds located in the Savannah (with larger
negative bias in watersheds with large rainfall amount) but tends to overestimate in all watersheds located in the Sahel
(with very large overestimation bias for the driest watershed) Therefore, GFS and IMERG Early have different bias

characteristics: whereas GFS outperforms IMERG Early in the Sahelian climate where well-organized convective systems dominate the monsoon, IMERG Early outperforms GFS in the Savannah and Guinea climate which are characterized by short-lasting and localized systems and wet land surface conditions. CHIRPS estimates are reasonably close to IMERG Final, indicating that the choice of reference product between CHIRPS and IMERG Final would not substantially affect the findings on the accuracy of GFS forecasts.

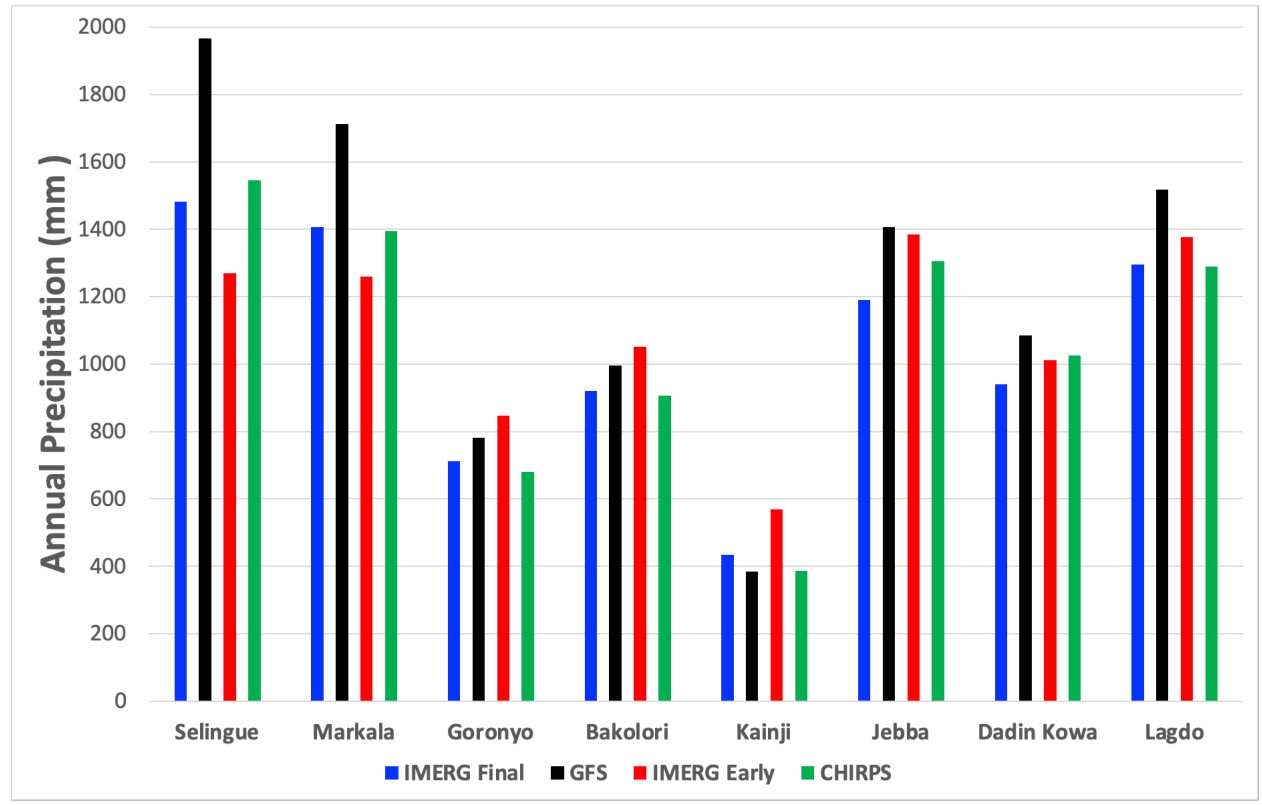

Figure 4. Sub-basin averaged annual precipitation (mm) for the period, 15 June 2019 to 15 June 2020, for each of the Niger's sub-basin, derived from the 1-day lead GFS forecast and different satellite precipitation products.

### 3.3 How well is the time series of daily precipitation forecasted?

Figures 5 and 6 present the time series of watershed-averaged daily rainfall, for the wet period June – October. According to IMERG Final, the temporal variability (as measured through coefficient of variation or CV) varies from 1.22 to 2.60. Validated against IMERG Final, the GFS tends to underestimate the temporal variability and particularly underestimate large spikes in rainfall, at almost all sites except at Kainji. The GFS's relatively better performance for

Kainji could be attributed to the watershed's large area that results in relatively smooth temporal variability. Both
IMERG Early and CHIRPS provide CV values that are very close to IMERG Final.

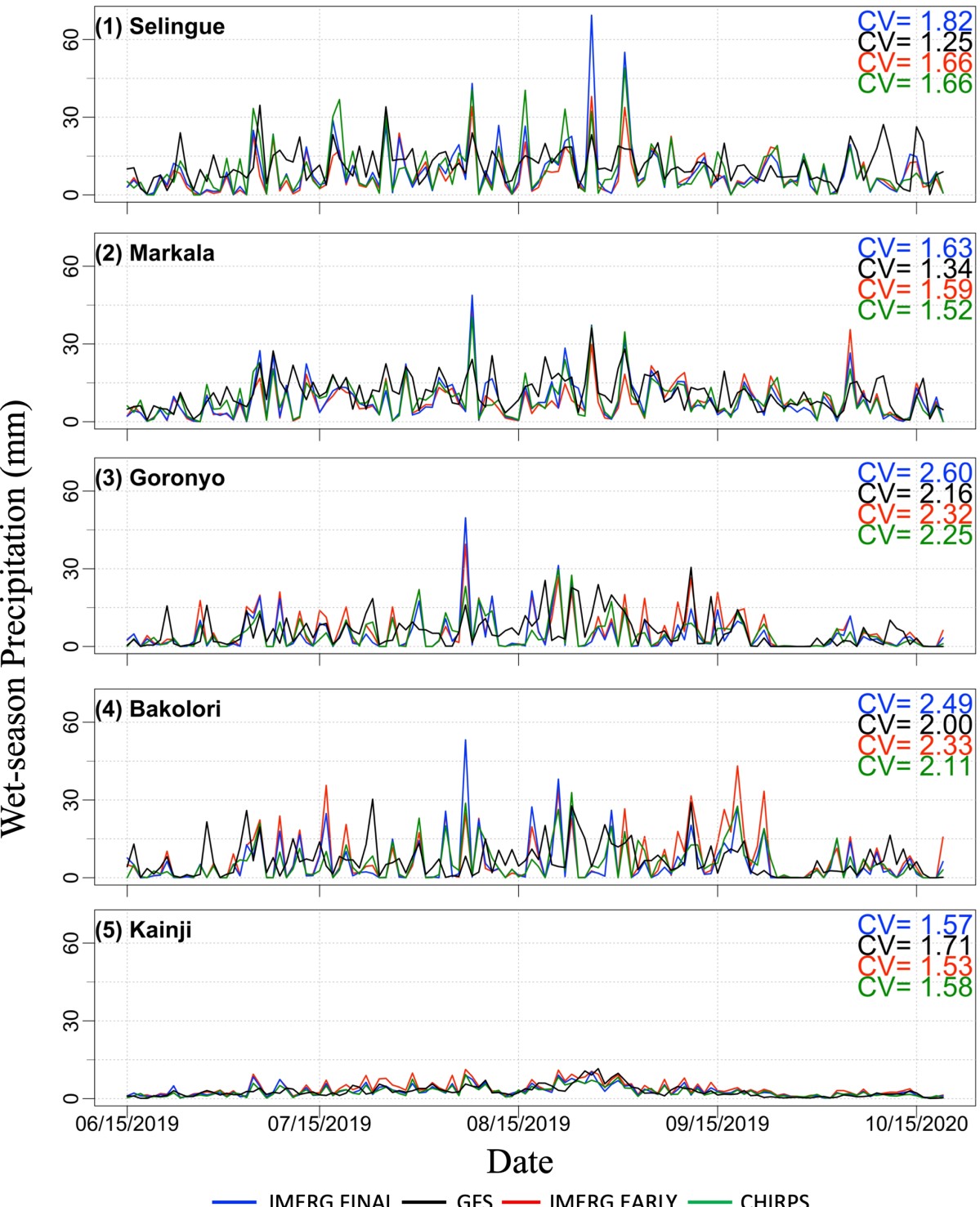

Figure 5. Time series of sub-basin averaged precipitation total (mm) for the wet period (June – September 2019 for all sub-basins, derived from various precipitation products, for five sub-basins. The Figure also shows the coefficient of variation (CV) as a measure of temporal variation.

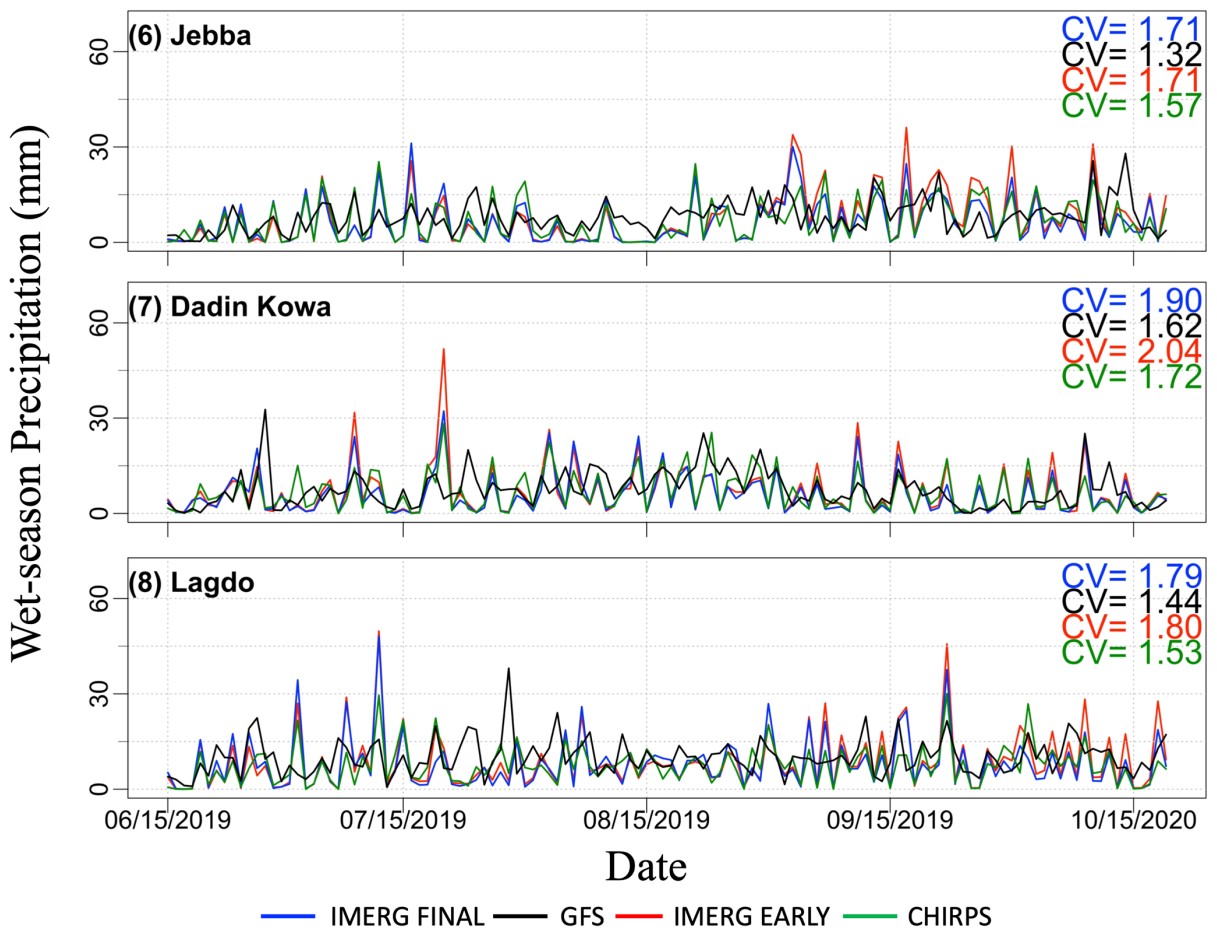

Figure 6. Same as in Figure 5 but for the remaining three watersheds.


Figure 7 displays the performance statistics of watershed-averaged daily rainfall (validated against IMERG Final) in
terms of Kling-Gupta Efficiency (KGE), Bias Ratio (BR), correlation (R), variability ratio (γ), and root mean square
error normalized by reference precipitation mean (NRMSE). First, the performance results for the 1-day lead GFS are
considered. The KGE scores are poor (0.3 < KGE < 0.5) for half of the watersheds considered (Selingue, Goronyo,
Bakolori, and Lagdo) and intermediate (0.5 < KGE < 0.75) for the remaining half watersheds (Markala, Kainji, Jebba,
and Dadin Kowa). The breakdown of the KGE scores (BR, R, and γ) reveals the key factors contributing to the KGE
estimates. The GFS tends to overestimate daily precipitation for most sub-basins, as BR is higher than one, except for
Kainji. The overestimation is particularly high for Selingue and Markala, where BR is 1.33 and 1.22, respectively.
The correlation coefficient between GFS and IMERG Final is mostly low (R < 0.60), and is particularly lower for
Bakolori (R=0.36) and Goronyo (R=0.43).  The variability ratio of GFS is mostly between 0.69 to 0.83 (except for
Kainji, where γ is 1.09), indicating that the GFS tends to give lower temporal variability of rainfall.  The NRMSE is
very high, ranging from 100% to 266%, and is particularly high for Goronyo (266%) and Bakolori (264%), which are
relatively small-sized watersheds.

Next, the performance of IMERG Early was examined with respect to IMERG Final, mainly to assess if it is possible
to use the near-real-time IMERG Early product to calibrate and improve the accuracy of GFS forecasts. The IMERG
Early performs much better with KGE values higher than 0.75 (except for Kainji where KGE is 0.69), correlation
higher than 0.90, and variability ratio close to the optimum value. The high performance of IMERG Early is due to its
similarity with the IMERG Final product, as the main difference between the two products is that IMERG Early,
unlike IMERG Final, does not use monthly rain gauge observations for bias correction. Such monthly bias correction
techniques would not alter the pattern and variability of IMERG Early compared to IMERG Final. Therefore, the
performance of IMERG Early should be evaluated using bias statistics, the other statistics (correlation and variability
ratio) are presented for completeness. IMERG Early overestimates rainfall in most watersheds in the range 11%
(Lagdo) to 28% (Kainji) except for two watersheds, where it slightly underestimates by 14% (Selingue) and 11%
(Markala). Comparison of the performance of GFS and IMERG Early indicates that both products have different bias
characteristics. In some watersheds (e.g., Kainji), GFS outperforms IMERG Early in terms of bias, whereas in other
watersheds (e.g., Markala), IMERG Early outperforms GFS.

CHIRPS was also compared with IMERG Final to assess how the use of different reference products may affect the
finding about the performance of GFS forecasts. The KGE scores of CHIRPS are higher than 0.75 in all cases,
indicating that CHIRPS and IMERG Final have comparable KGE performance. Therefore, the performance of GFS
is expected to be about the same even if the reference product used this in this study (IMERG Final) changes to
CHIRPS.





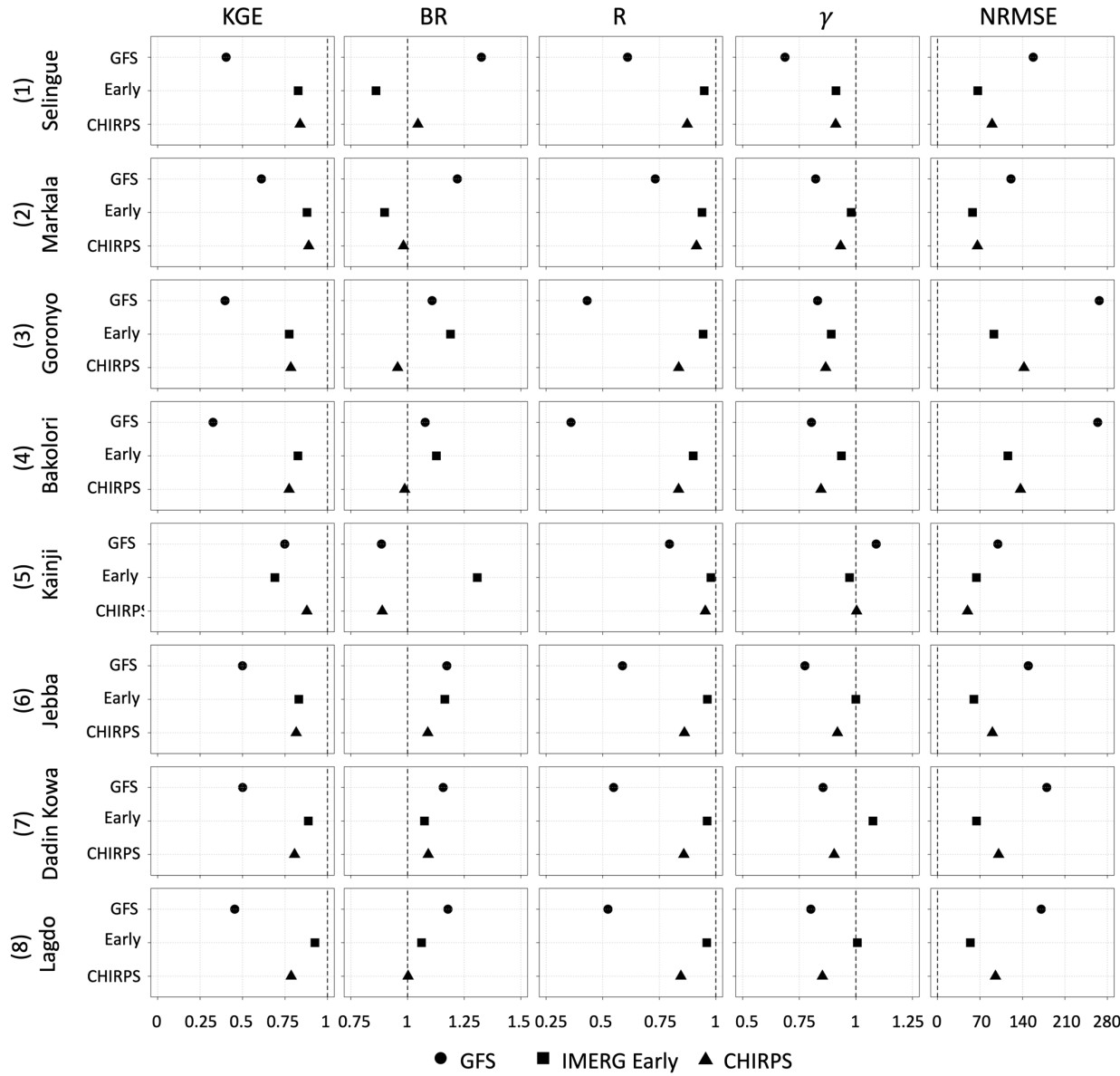

Figure 7. Summary of performance statistics (Kling-Gupta Efficiency KGE, Bias Ratio BR, correlation R, variability ratio $\gamma$, and root mean square error normalized by reference rainfall [%], for the 1-day lead time GFS forecasts and other satellite products. The time period considered was June 15, 2019 – June 15, 2020.






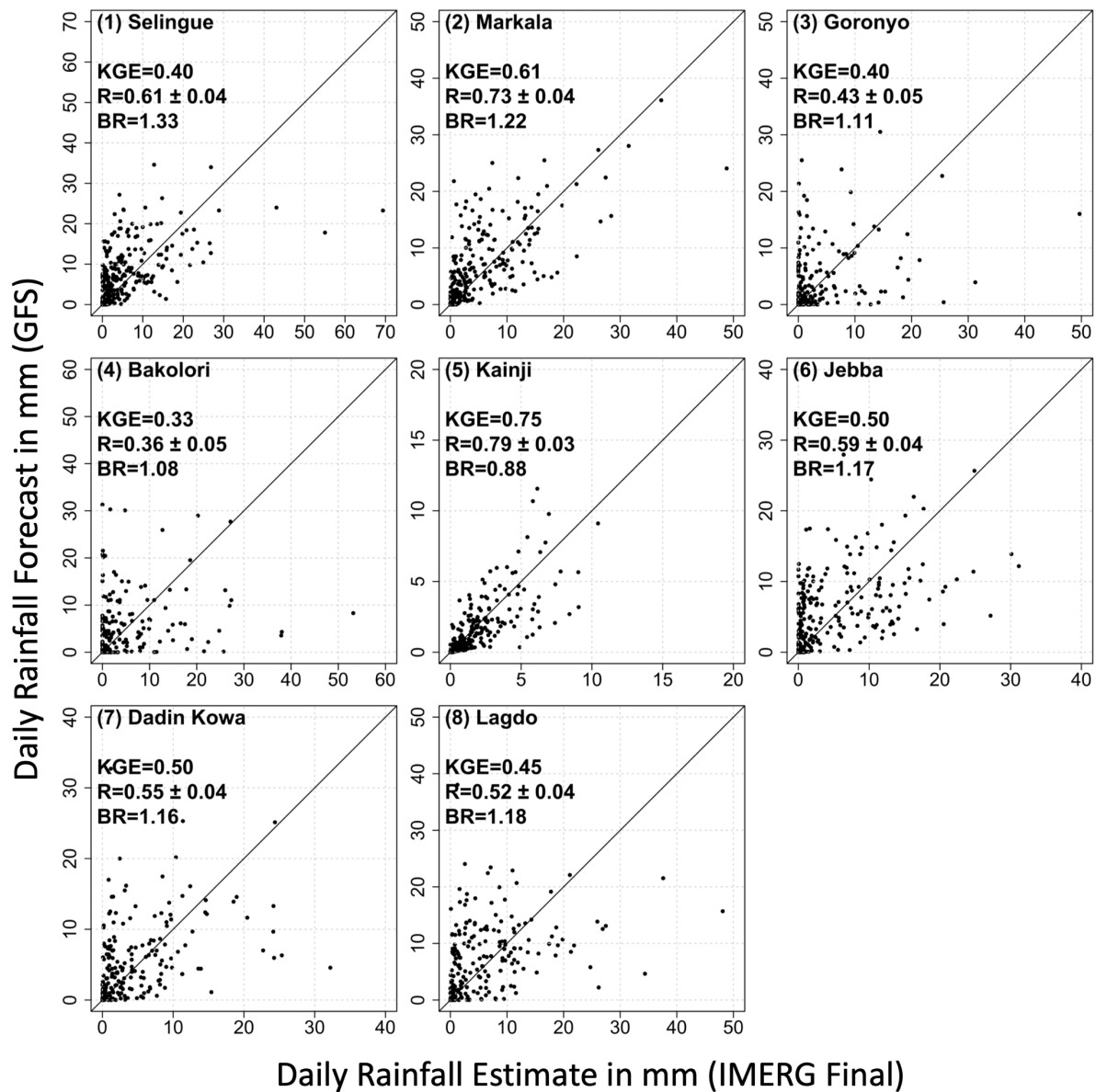

Figure 8. Scatterplot of watershed-averaged daily precipitation forecast obtained from 1-day lead GFS forecasts against corresponding values from IMERG Final.




**3.4 Dependence of Forecast Performance on Precipitation Rate**

Figure 8 presents the scatterplot of 1-day lead GFS forecasts and IMERG Final at daily and watershed-average scales.

The performance of GFS varies between watersheds. In the Markala and Kainji watersheds, GFS forecasts agree well

with IMERG Final at almost all rain rates. In the Selingue watershed, GFS agrees well with IMERG Final for rain
rates under 30 mm/day, but GFS substantially underestimates all rain rates above 30 mm/day. In the remaining five
watersheds, GFS has poor performance, replete with large scatter, high false alarm, and large underestimation bias of
heavy rain rates.

**3.5 Dependence of Daily Forecast Performance on Lead Time and Spatial Scale**
In order to assess the effect of various lead times and spatial scales on forecast performance, we obtained daily GFS
forecasts at various lead times (1-day, 5-day, 10-day, and 15-day), and aggregated the forecasts at spatial scales from
0.25° to coarser scales (0.5°, 0.75°, and 1°) by averaging grids. The purpose of degrading the resolution is to determine
at which resolution the forecasts have acceptable performance. The KGE value at each spatial resolution was
calculated in the following steps: (i) average the data at the required spatial resolution, (ii) extract pairs of data (one
from IMERG Final, and the other from GFS), (iii) concatenate the pairs to form one large series of data, and (4)
compute a single KGE from this data series. The resulting KGE values are shown in Fig. 9.

With regard to the effect of spatial scales, the KGE at the GFS native resolution (i.e. 0.25°) is very low. As the spatial
scale increases, KGE increases, as expected. For instance, for Markala watershed KGE increases from 0.27 (0.25°) to
0.40 (1°) for a 1-day lead. This indicates that the variation in KGE values between the watersheds could be partly
explained by the watershed size. For example, based on Fig. 5, the KGE for the 1-day lead daily GFS forecast was the
highest for the largest Kainji watershed (watershed area of 1,464,092 $Km^2$) and the lowest for the smallest Bakolori
watershed (4,887 $Km^2$). With regard to the effect of lead time for daily forecasts, KGE decreases significantly as lead
time increases. For instance, for Markala watershed and a grid size of 1°, KGE decreases from 0.40 (1-day lead) to
0.21 (15-day lead).


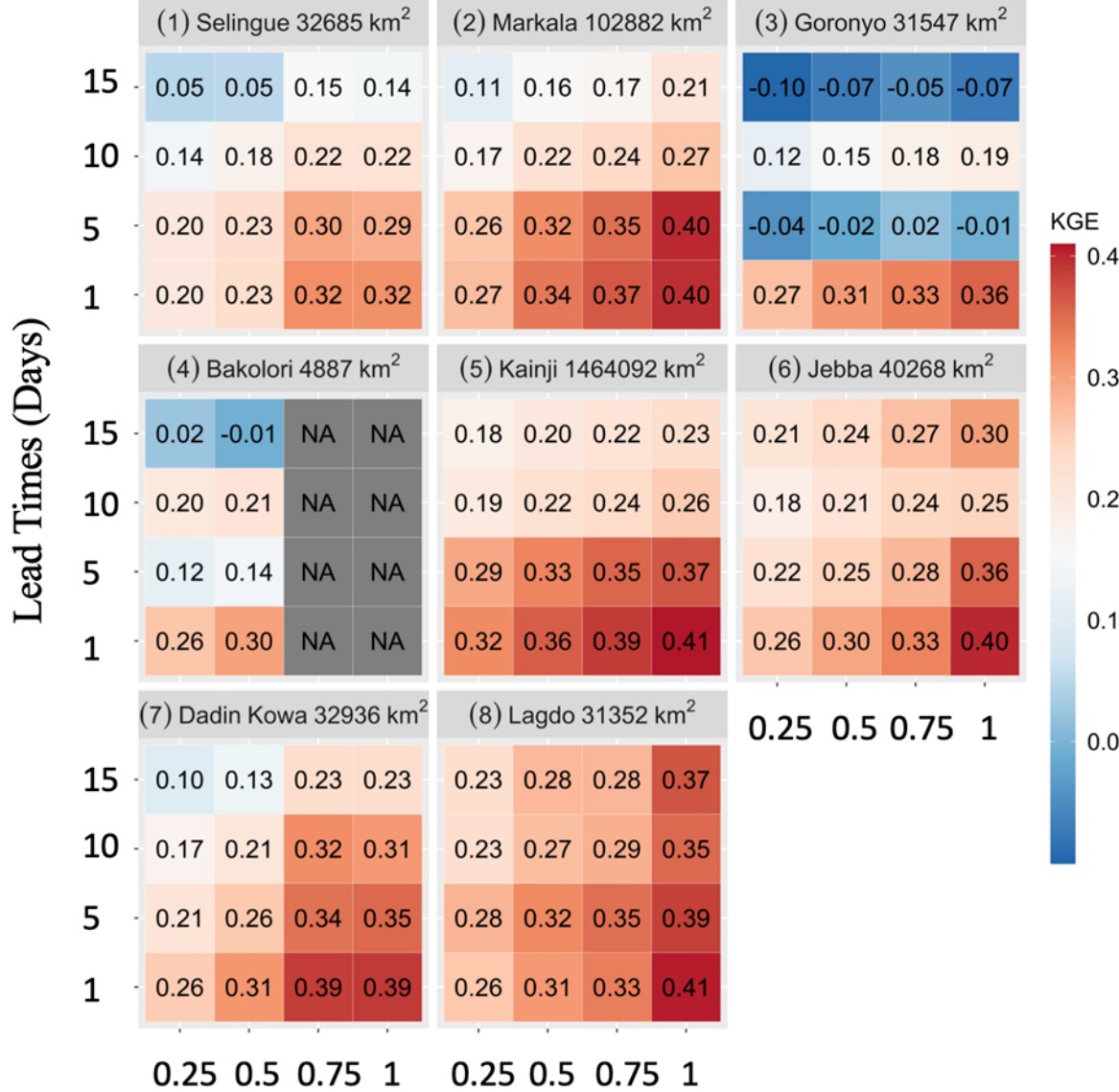

Figure 9. Kling-Gupta Efficiency (KGE) for daily precipitation of GFS as a function of lead time (1-day, 5-day, 10-day, and 15-day) and spatial scale (0.25°, 0.50°, 0.75°, 1.0°). The dam names and corresponding watershed areas are given in the titles.


**3.6 Effect of Temporal Aggregation Scale on Forecast Performance**

To assess the effect of temporal aggregation scale, we obtained the 1-day total, 5-day total, 10-day total, and 15-day

total GFS precipitation forecasts. These multi-day forecasts are constructed by combining multiple lead-time forecasts.

For instance, the 5-day total forecast is obtained by adding the 1-day lead, 2-day lead, 3-day lead, 4-day lead, and 5-
day lead daily forecasts. Figure 10 presents the KGE values for GFS forecasts over different temporal aggregation
scales, and different grid sizes. Temporal aggregation substantially increases KGE at all spatial scales. For example,
at the grid size of 1° over Markala watershed, the KGE values jump from 0.40 at daily timescale to 0.73 at 15-day
total timescale.

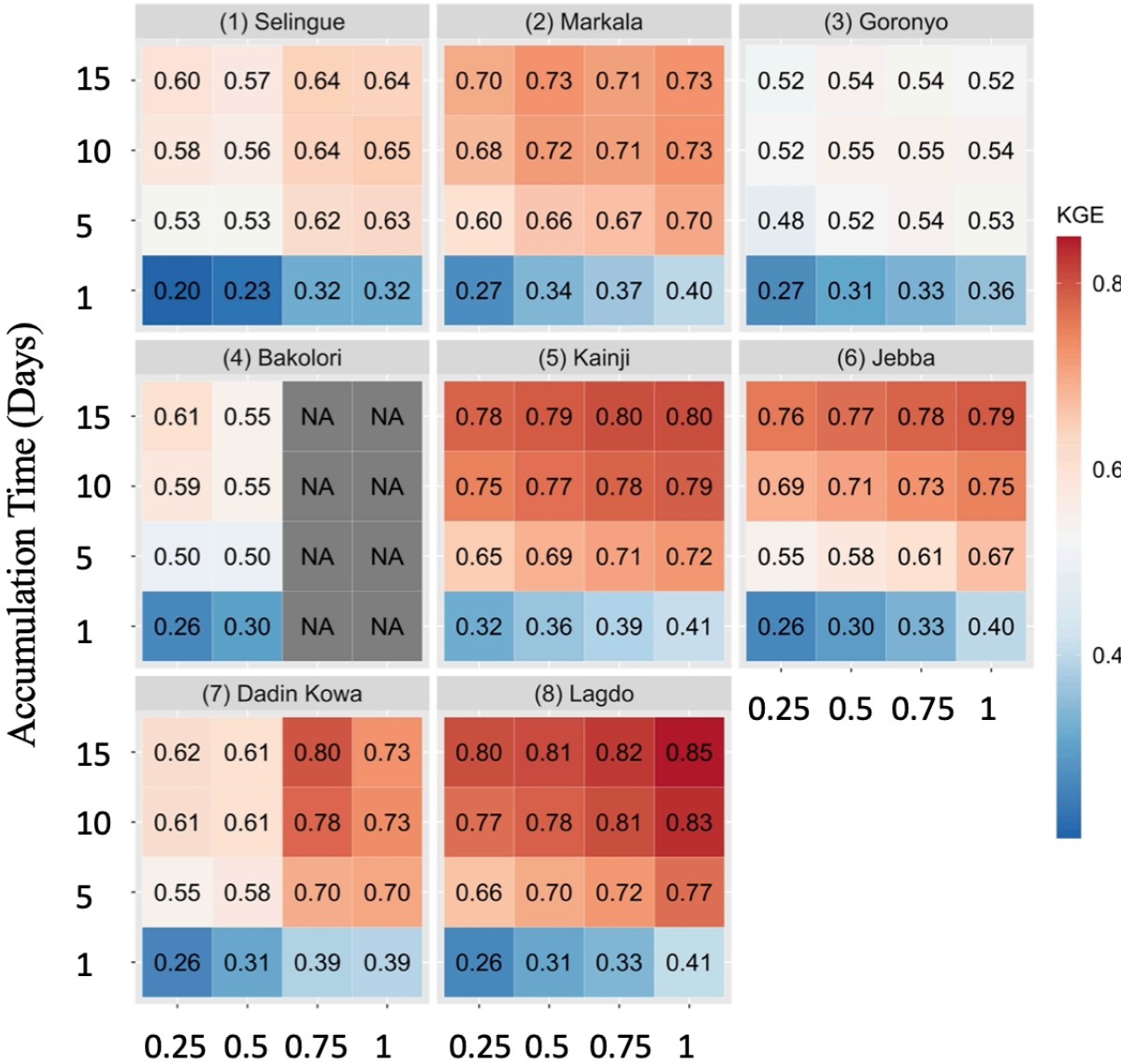

Figure 10. Kling-Gupta Efficiency (KGE) of GFS as a function of accumulation time scale (1-day, 5-day, 10-day, and 15-day) and spatial scale (0.25°, 0.50°, 0.75°, 1.0°).


In Figure 11, we show the performance statistics of GFS for 15-day accumulated watershed-averaged rainfall forecast.
The KGE values are intermediate (0.5 < KGE < 0.75) for four watersheds and good (KGE > 0.75) for the remaining
four watersheds. Analysis of the components of KGE reveals that the improvement of KGE at longer timescales comes
as a result of improved correlation and variability ratio. At the 15-day accumulation timescale, IMERG Early estimates
have less bias than GFS at all watersheds, except at Kainji watershed. Figure 12 presents the scatterplot of 15-day
accumulated GFS forecast vs IMERG Final. In general, the GFS estimates perform well for low to moderate rain rates,
but tend to overestimate higher rain rates. This is consistent with Wang et al. (2019) who reported the difficulty of
capturing the magnitude of high rain rates in GFS model.

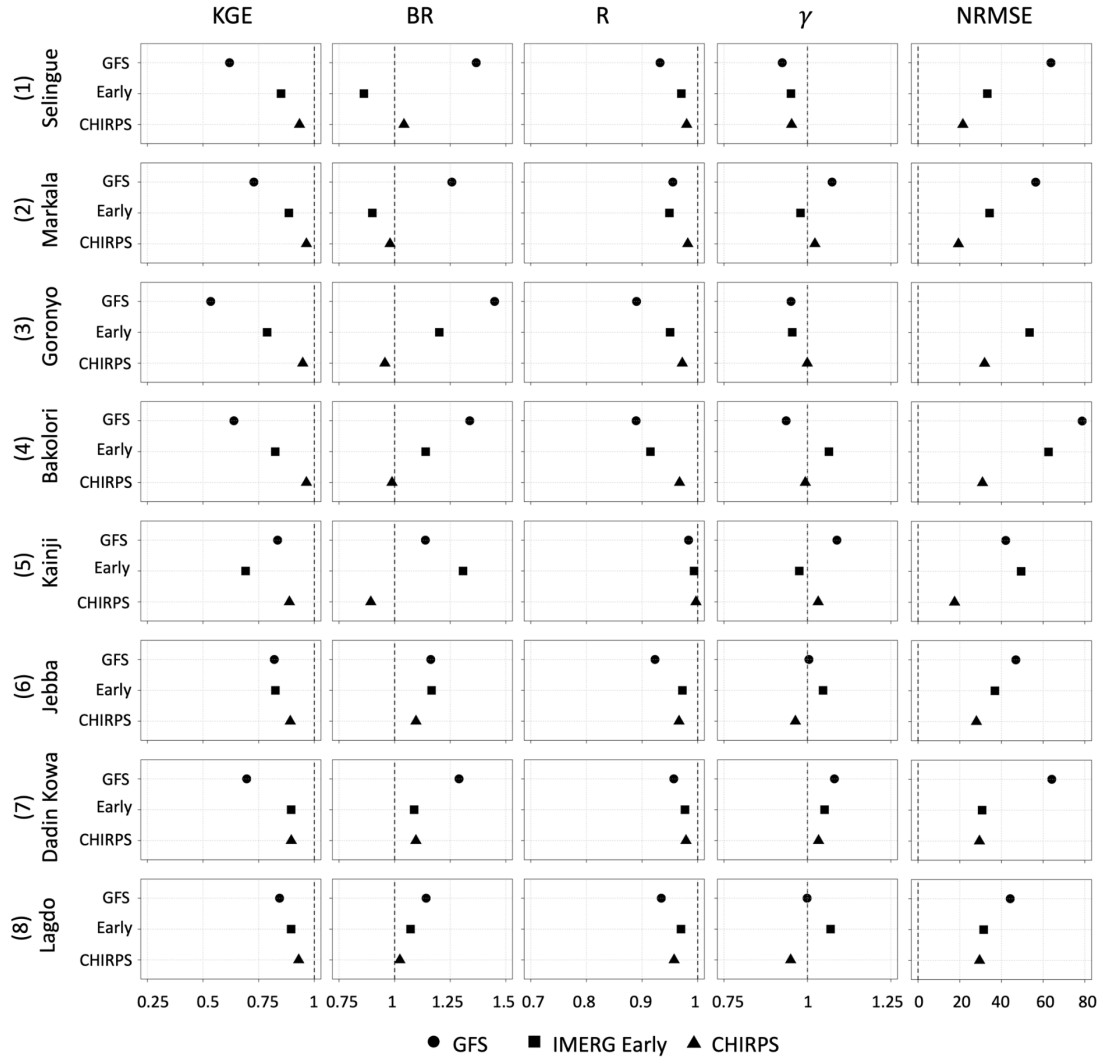

Figure 11. Summary of performance statistics (Kling-Gupta Efficiency KGE, Bias Ratio BR, correlation R, variability ratio γ, and root mean square error normalized by reference rainfall [%], for the 15-day accumulated GFS forecast and other satellite products.

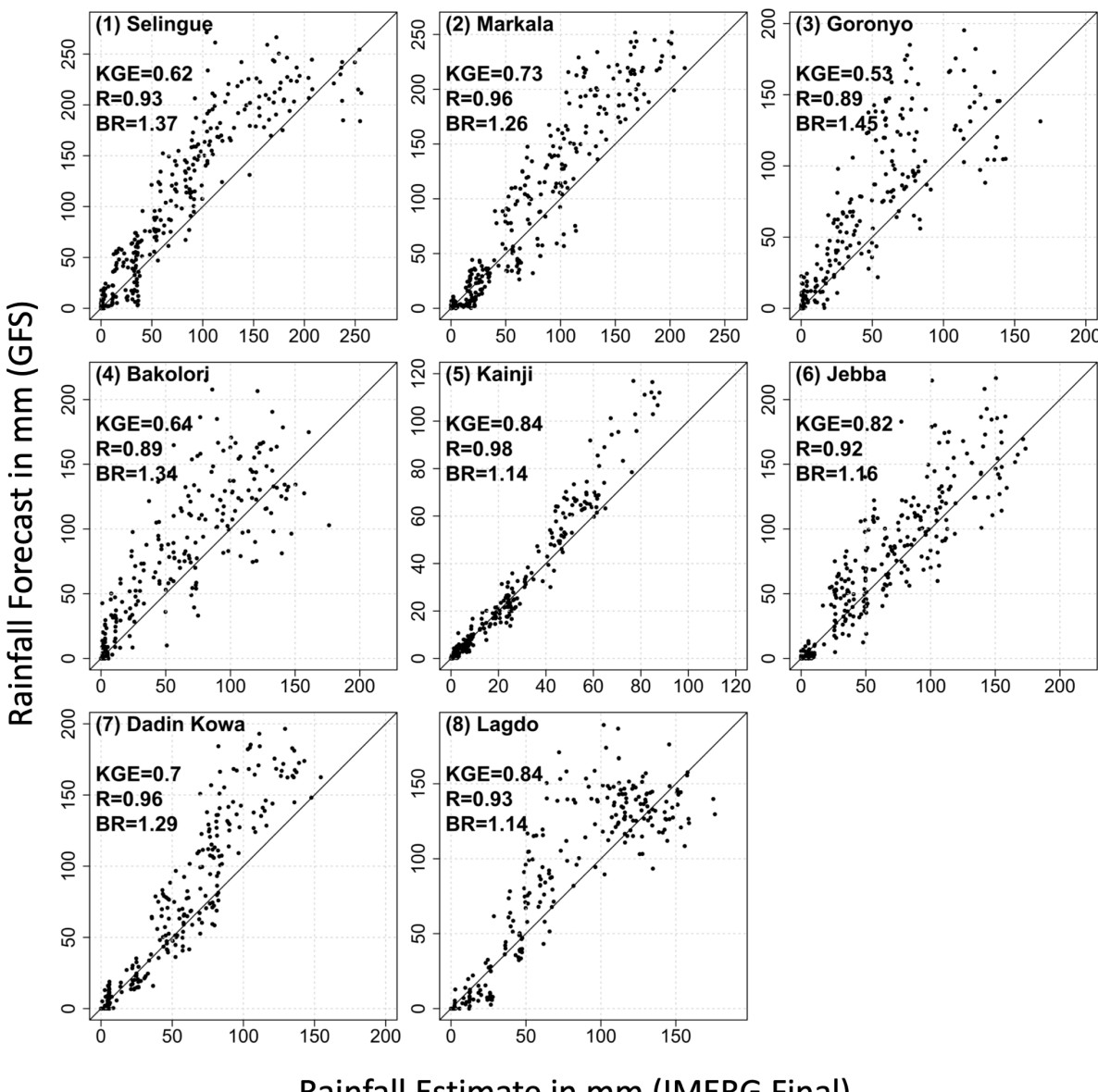

Figure 12. Scatterplot of watershed-averaged 15-day accumulated precipitation forecast obtained from GFS forecast against corresponding values from IMERG Final.


## 4.  Conclusions

This study has evaluated the accuracy of medium-range (1-day to 15-day lead time) forecasts available from the Global

Forecast System (GFS), for the watersheds of large dams in the Niger river basin. Despite the limited temporal

coverage, some consistent features emerged from this evaluation. The accuracy of GFS forecast depends on climatic
regime, lead time, accumulation timescale, and spatial scale. With regard to the role of climatic regimes, the GFS
forecast has large overestimation bias in the Guinea (wet climatic regime), moderate overestimation bias in the
Savannah (moderately wet climatic regime), but has no bias in the Sahel (dry climate). With regard to lead time, as
the lead time increases, the forecast accuracy decreases. Averaging the forecasts at coarser spatial scales leads to
increased forecast accuracy. For daily rainfall forecasts, the performance of GFS is very low (KGE < 0.32) at almost
all watersheds except at Markala (KGE = 0.44) and Kainji (KGE = 0.68), both of which have much larger watershed
areas compared to the other watersheds. Averaging the forecasts at longer time scales also leads to increased forecast
accuracy. For 15-day rainfall accumulation timescale, the KGE values are either "intermediate" (i.e., 0.50 ≤ KGE ≤
0.75) for half of the watersheds (Selingue, Goronyo, Bakolori, and Daddin Kowa) or "good" (i.e., KGE ≥ 0.75) for
the remaining half (Markaa, Kainji, Jebba, and Lagdo). With regard to the effect of rainfall rate, the 15-day
accumulated GFS forecasts tend to perform better for low to medium rain rates, but contain large overestimation bias
at high rain rates.

The performance statistics of GFS indicate the need for calibrating GFS forecasts in order to improve their accuracy.
Post-processing calibration of GFS forecasts requires the use of "relatively better performing" and "available in near-
real-time" independent rainfall observations to correct real-time dynamical GFS model forecasts. This study has
compared the performance of IMERG Early satellite rainfall products with the performance of GFS in terms of bias.
In the Guinea and Savannah regions, IMERG Early outperforms GFS in terms of bias, while in the dry Sahel region,
IMERG Early is outperformed by GFS.

We acknowledge that the reference dataset used in our evaluation (i.e., IMERG Final) has its own estimation errors.
We conducted additional assessment to evaluate the performance of IMERG Final with respect to another independent
and high-quality (i.e. satellite-gauge merged) rainfall product (i.e. CHIRPS). Our results show that IMERG Final and
CHIRPS have similar rainfall characteristics, indicating the robustness of IMERG Final.

Overall, we conclude that the GFS forecasts, at 15-day accumulation timescale, have acceptable performance,
although they tend to overestimate high rain rates. The shorter the time scale, the lower is the GFS performance. Given
that IMERG Early outperforms GFS particularly in the wet region of the Niger, we recommend testing the suitability
of IMERG Early to serve as input into post-processing of GFS in order to improve the accuracy of GFS forecasts.
Possible post-processing techniques that could be explored include: simple bias (multiplicative) correction
(Gumindoga et al. 2019), multi-resolution bias correction through wavelet analysis (Xu et al. 2019) or empirical mode
decomposition method (Wang et al. 2020, Prasad et al. 2019), and Artificial Intelligence-based methods such as Feed
Forward Neural Network (Cloud et al. 2019), Support Vector Machine ((Du et al. 2017; Yu et al. 2017), and Adaptive
Neural Fuzzy Inference System (Jehanzaib et al. 2021).





















**5. Data and Code Availability**

We acknowledge the National Center for Atmospheric Research (NCAR) for providing public access to the GFS rainfall forecast data products (https://rda.ucar.edu/datasets/ds084.1/), NASA for providing public access to IMERG Final and IMERG Early rainfall data products (https://disc.gsfc.nasa.gov), and the University of California Santa Barbara's (UCSB) Climate Hazard's group for providing public access to CHIRPS rainfall data (https://www.chc.ucsb.edu/data).

**6. Author Contribution**

H. Yue: data processing, data analysis, and manuscript preparation; M. Gebremichael: project oversight, method design, contribution to manuscript text; V. Nourani: method design, contribution to manuscript text.

**7. Competing Interests**

The authors declare that they have no conflict of interest.

**8. Acknowledgement**

We acknowledge funding support from NASA Precipitation Measurement Mission through Grant # 80NSSC19K0688.

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
