# Peer review of "Performance of the Global Forecast System's Medium-Range"

_Hydrology and Earth System Sciences, 2021_

## Author Comment (AC2)

**Response to Referee Comment #2**

Thank you for detailed comments. We have now revised the manuscript accordingly. Please find below our itemized responses.

**Main Comments:**

1.  In the abstract, the authors mentioned that application of post-processing techniques involving near-real time satellite rainfall products could improve the accuracy of the GFS forecasts. However, no supporting analysis was presented in this manuscript for this statement. What is the basis of this statement? If it is based on some reference study, please provide details of those studies. If not, then please include appropriate analysis results in support.

    **Response:** We agree that we have not provided supporting analysis, and therefore have replaced the statement "*The accuracy of GFS forecasts could be improved by applying post-processing techniques involving near-real time satellite rainfall products.*" with "*We recommend exploring appropriate post-processing calibration techniques, that use near-real time products, such as, IMERG Early, to improve the performance of GFS, particularly at shorter time scales.*"

    The reason for our recommendation is that IMERG Early outperforms GFS in most cases, and therefore, it can be used to calibrate GFS. The IMERG Early outperform GFS in Wet Guinea and Savannah regions in terms of bias, and the spatial structure of IMERG Early is the same as IMERG Final – as the main difference between the two products in the inclusion or exclusion of rain gauge data which affects primarily the bias. However, it is not clear what kind of post-processing technique is more appropriate.

2.  In section 3.5, the authors attempted to correct for biases in the IMERG Early precipitation products with climatological input and concluded that climatological bias correction is not effective for IMERG Early products. However, there was no significant relevance of this section to this manuscript. What was the purpose of presenting these analyses in this manuscript?

    **Response**: We agree that it is not meaningful to present results that do not improve performance. Hence, we have removed the climatological bias correction from our evaluation. In the Conclusions section (see last paragraph), we have recommended the need for identifying appropriate bias calibration techniques, and suggested some potential methodologies that could be explored.

3.  The authors stated in conclusion that the GFS forecasts are almost unbiased at low to medium rain rates. However, it is unclear which GEFS forecast product they are referring to by this statement. It is my assumption that they are referring to the 15-day accumulated precipitation here which should be clearly stated to avoid confusion.

**Response**: Yes, we have now added the qualifier '15-day accumulated'.

4. The authors claimed in conclusion that the use of IMERG Early to calibrate GFS would improve GFS forecast quality, however, there was no evidence presented to support this statement. Did the authors performed any analysis that has not been included in this manuscript and reached to this conclusion? If so, they should include the results from that analysis to support their claim.

   **Response**: We have now removed this statement, as we have not provided supporting evidence. In addition, we realized that a suitable post-processing technique needs to be developed to take advantage of the (relatively better) performance of IMERG Early. We added a recommendation that reads "*We recommend identifying suitable post-processing calibration techniques, through the use of near-real time products, such as, IMERG Early, that could improve the performance of GFS, particularly in the wet Guinea and Savannah regions. Possible calibration methods that could be explored include: simple bias (multiplicative) correction, multi-resolution bias correction through wavelet analysis wavelet analysis or empirical mode decomposition method, and Artificial-based methods such as Feed Forward Neural Network (FFNN), Support Vector Machine (SVR), and Adaptive Neural Fuzzy Inference System (ANFIS).*"

5. Since the area studied involves multiple dams used for water supply, irrigation, hydropower, etc., I would have loved to see some event specific results to demonstrate GFS products' ability to forecast significantly dry and wet event which is very significant for reservoir management.

   **Response**: We think this could be extracted from our figures. Figures 5 and 6 show the performance of GFS for each day, and each reservoir location. Figures 8 and 12 show the performance of GFS as a function of rain rate, for two time scales, and for each reservoir location.

**Minor Comments:**

1. Axes should be labelled appropriately to convey that these are 15-day accumulated precipitation amounts instead of "daily".

   **Response**: We have removed 'Daily' from the labels. The timescale is shown in the figure caption.

2. Page 2, Line 13: "with annual rainfall of rainfall" should be rewritten as "with annual rainfall of".

   **Response**: The text has been removed during abstract re-writing.

3. Page 3, Line 29: Please check the reference "Saha et al. 2011". Should this be "Saha et al. 2014"?

   **Response**: We agree. It is fixed now.

4. Page 4, Lines 76-78: Please check this sentence and rewrite.
   **Response:** It has been re-written now.

5. Page 5, Line 106: "Hybrid eddy-diffusivity mass-flux (EDMF)".
   **Response**: It is fixed now.

6. Page 6, Line 128: "The Climate Hazard's group Infrared Precipitation (CHIRP) and with Stations (CHIRPS)" should be rewritten as "The Climate Hazards Group InfraRed Precipitation with Station data (CHIRPS)".
   **Response**: We have fixed it now.

7. Page 7, Line 164: In terms of the size of the watershed, smallest among the study dams is the Bakolori Dam. However, the author said it is the Markala Dam on this line. The unit of the watershed area should also be changed to "km2" from "m2".
   **Response**: We have fixed this now.

8. Page 10, Line 209: "CV is the coefficient of variation" instead of "CV is the coefficient if variation"
   **Response**: It is fixed now.

9. Page 23, Lines 393-394: Please rewrite this sentence.
   **Response**: It has been re-written.

10. Page 26, Line 432: "some consistent features emerged" instead of "some consistent featured emerged".
    **Response**: It has been fixed now.

---

## Author Comment (AC3)

**Response to Community Comment #1**

**Overview Comment:**

I enjoyed reading this interesting work that has the potential to contribute to reservoir operations in the Niger Basin. I think the topic of the manuscript fits well to the journal. The use of language needs minor edits and the structure could be improved. I listed my main and minor comments below, that I think will improve the flow, the clarity and the significance of the manuscript.

> **Response**: Thank you very much for your great inputs. We are glad to hear that you enjoyed the work. We agree that the work is important in utilizing forecasts to improve reservoir operation. We have tried our best to address all your comments as we revised the manuscript. Please see below on our itemized response.

**Main Comments:**

1) Abstract: It is not clear which rainfall dataset is used as reference for performance analysis of the GFS forecast. Only later in the text it is mentioned that IMERG Final is the reference dataset. Please provide the numerical values for the performance statistics. For example, the numerical values for the overestimation, underestimation, large random errors, high false alarm etc should be provided. Moreover, a statement that other satellite products are also compared should be provided in abstract.

> **Response**: We have now included (1) a statement indicating that IMERG Final is used as a reference, and (2) a statement indicating that other satellite data products are also compared. We prefer not to provide numerical values for the error characteristics, as they vary highly depending on watershed, climatic regime, lead-time, averaging timescale, averaging spatial scale, and rainfall rate.

2) Abstract: Last sentence: It is not clear whether the authors performed an analysis to support this statement. If yes, this statement should be supported with the method and findings utilized, otherwise it is a general statement and should be removed from abstract.

> **Response**: We agree that we have not provided supporting analysis, and therefore have replaced the statement *"The accuracy of GFS forecasts could be improved by applying post-processing techniques involving near-real time satellite rainfall products."* with *"We recommend exploring appropriate post-processing calibration techniques, that use near-real time products, such as, IMERG Early, to improve the performance of GFS, particularly at shorter time scales."*
>
> The reason for our recommendation is that IMERG Early outperforms GFS in most cases, and therefore, it can be used to calibrate GFS. The IMERG Early outperform GFS in Wet Guinea and Savannah regions in terms of bias, and the spatial structure of IMERG Early is the same as IMERG Final – as the main difference between the two products in the

inclusion or exclusion of rain gauge data which affects primarily the bias. However, it is not clear what kind of post-processing technique is more appropriate.

3) I think Section 2.Data and Methodology should be divided into two sections namely "2. Study Area and Datasets" and "3. Methodology". Lines 141-196 should move to the 2.1. Study Area section. Current Sections 2.1-2.3 should move to new "2.2. Datasets" section. Current Section 2.4 should move to "3.Methodology" section. This section should also include other data processing methods used in the study such as scale matching between products, basin-scale conversion etc. as well as study time period.

> **Response**: We have re-arranged the sections as follows. We have created a separate section labeled "Study Region" and moved Lines 141 – 196 to the Study Region section.  In the 'Evaluation' Section, we have added the data processing methods.

4) IMERG Early Cal: This product is not shown in Figure 3 and shown in Figure 4 but not discussed in text. The reader has no information about this product until Section 3.5. To eliminate this confusion, please discuss the motivation for producing this rainfall dataset and the methodology for adjusting IMERG Early using IMERG Final in the Methodology section (Lines 347-352 in the manuscript). It may be worthwhile to indicate that the advantage of simple climatological correction for IMERG Early would be the shorter lag time (a few hours) compared to IMERG Final (3.5 months).

> **Response**: We have decided to remove the 'IMERG Early Cal' dataset from our evaluation (as also suggested by another Reviewer).  The climatological bias correction approach used to generate the 'IMERG Early Cal' dataset did not really improve performance compared to IMERG Early.

5) Section 3.4: This section shows the scatterplots comparing correspondence between daily rainfall events between IMERG Final and GFS. Therefore, it is not related to uncertainty but a different way of comparing daily events. Section 3.4 can therefore be merged with Section 3.3.

> **Response**: Yes, however, we formed our different sections based on the different ways of comparisons.

6) Section: 3.6: Please discuss the methodology for changing spatial scale of the products in the Methodology section. Moreover, indicate the reference rainfall product used in this section (IMERG Final). It may be helpful for the reader to include the watershed area next to each watershed name in Figure 9. Also somewhere in the manuscript, the number of rainfall product grids over each dam watershed should be provided.

> **Response**: We have included the methodology in the 'Evaluation' Section.  We have added the watershed areas.

7) I suggest that the title of the manuscript should be modified to include the use of satellite rainfall products in the comparison. For example something similar to "Performance evaluation of the Global Forecast System's Medium-Range Precipitation Forecasts in the Niger River Basin using multiple satellite-based products."

> **Response**: Yes, this is a better title. We have used it.

8) In conclusion section, a discussion on the study findings for dam operation would be beneficial for the reader since the focus is on dam watersheds (for example the impact of change in lead time performance in dam operation).

> **Response**: Yes, this is an important issue. However, it is difficult for us to state the implication of the forecast accuracy on reservoir operation at this stage, simply because the relationship between forecast accuracy and dam operation is not well-known. This could be a good topic for future research – thank you for the idea.

**Minor Comments:**

Line 37: Typo "Nige" should be corrected "Niger"
> **Response**: Corrected.

Line 36: Figure 1: Figure should appear in the same or next page of the first referral.
> **Response:** Done.

Lines 56-59: This last sentence should move to the next paragraph.
> **Response**: Done.

Line 61: A reference (Huffman etal.) to IMERG product should be provided early in this paragraph.
> **Response**: Done.

Line 76: Typo, please correct "IMERG Fsatellite gauged)"
> **Response**: Corrected.

Line 80: Please provide temporal and spatial resolutions in parenthesis.
> **Response**: We have added this information.

Line 82: Replace "motivate" with "motivated"
> **Response**: The word has been removed due to rearranging.

Line 106: Typo. "mass-flus"
> **Response**: Fixed.

Line 118: Remove "Earth Data"
> **Response**: Word has been removed.

Line 164: Unit is missing for Markala Dam watershed size.
> **Response**: Added.

Line 199: Replace "previous" with "upstream"
> **Response**: Replaced.

Table 1: Is there a source for this information?
    **Response**: Yes, source has been added.

Figure 1: Please overlay GFS, IMERG and other satellite-product grids on this figure as a reference and to better understand dataset scale in comparison to dam watershed scale.
    **Response**: We tried this (see below) but it did not come out well. So, we opted to keep the original figure.

[Figure]

Line 209: Replace "coefficient if variation" with "coefficient of variation"
    **Response**: Replaced.

Line 207: Coefficient of variation is used in the modified KGE measure proposed by Kling et al (2012) and generally denoted by KGE'. Therefore, please include this information in description of KGE used in this study.
    **Response**: It is now included.

Lines 211-212: I do not recall this classification by Kling et al. (2012). Please check to make sure correct citation is provided for this KGE classification.
   **Response**: We have replaced it by the correct reference.

Lines 215-216: The following information is also important and can be included here: R measure is important in reproducing the temporal dynamics.
   **Response**: We have added it.

Figure 2: It will help readability if horizontal lines are drawn to represent the regions (4,8,12 degrees) as shown in Figure 1.
   **Response**: We have drawn the requested horizontal lines.

Figure 3: Please include which year this graph represents in the caption or text.
   **Response**: Temporal period is provided in the caption.

Line 254: Please provide the methodology used to calculate watershed-averaged rainfall in the methodology section. How many rainfall grids represent each basin etc.
   **Response**: We have added the methodology in Section 2.5 (Evaluation). The watershed areas are already given in Table 21.

Figure 4: Check spelling for Goronye sub-basin throughout the manuscript, for example in Figure 1 it is Goronyo.
   **Response**: We have done this.

Figure 7: Please provide time period information in caption. Color coding of the markers, similar to Figure 5, will improve readability.
   **Response**: We have added the time period information in the caption. We are trying to avoid color as much as possible (to cut publication cost). We thought that the different positions of GFS and satellite products will make it easier to read -

Line 327: I suggest modifying the sentence: The overestimation by IMERG Early is particularly…
   **Response**: The sentence was removed due to re-writing.

Figure 10: This figure should come after first referral in the text (Section 3.7).
   **Response**: We have moved the figure.

Lines 393-394: Check grammar.
   **Response**: We have fixed this.

Figure 12: "daily" should be removed from x and y axis titles.
   **Response**: Done.

Abstract: Include the findings from Sections 3.6 and 3.7 in the abstract.

**Response**: We have re-written the abstract to include these findings.

Line 432: Typo "featured emerged" replace with "features emerged"

**Response:** Typo fixed.

Lines 449-450: I did not see a section on calibration of GFS using IMERG-Early. Please clarify or remove this sentence

**Response**: We have removed this sentence.

---

## Author Comment (AC4)

**Response to Referee Comment #1**

Thank you for your insightful comments, which have helped strengthen the manuscript. We have revised the manuscript accordingly, and please find below our itemized responses.

In the conclusions, the authors states that "The use of IMERG Early to calibrate GFS would improve GFS in terms of correlation and variability, but not in terms of bias". How do you come to this conclusion? Just via comparing the performance of GFS forecasts and IMERG Early products? I think it is inadequate. I strongly suggest that the authors should add some more analysis on the comparison of performance between raw GFS forecasts and calibrated GFS forecasts by IMERG Early products. In addition, the authors also evaluate the performance of some other Satellite Precipitation Products, such as CHIRP, IMERG Early and IMERG Early Cal, against the IMERG Final rainfall products. However, I do not understand why you evaluate these products? You evaluated these products but did not do any analysis on using these products to improve the GFS forecasts.

**Response**:
- We acknowledge the importance of comparing the performance of raw forecasts with calibrated GFS forecasts. However, this would require developing new appropriate methodologies, which is outside the scope of this study. We have added a new paragraph (paragraph 4 of the Conclusions Section) indicating the need for developing such a methodology, and suggested potential methodologies that could be explored. We plan to pursue this in future research.
- As far as the purpose of evaluating the performance of different datasets (CHIRP, IMERG Early, and IMER Early Cal) is concerned:
    - **IMERG Early Cal**: The climatological bias correction approach used to generate the 'IMERG Early Cal' dataset did not improve performance compared to IMERG Early. Thus, we have decided to remove the 'IMERG Early Cal' dataset from our evaluation (as suggested by the Reviewer in a separate comment below).
    - **CHIRPS**: In this study, IMERG Final has been used as a reference to evaluate the performance of GFS forecasts. We conducted additional assessment to evaluate the performance of IMERG Final with respect to another independent and high-quality (i.e. satellite-gauge merged) rainfall product (i.e. CHIRPS). Agreement between the reference (IMERG Final) and CHIRPS would indicate that the IMERG Final estimates are robust. We have added a new paragraph in the Conclusions section (paragraph 2) as well as additional texts in Section 2.3 to clarify this.
    - **IMERG EARLY**: Post-processing calibration of GFS forecasts (in order to improve the accuracy of GFS) requires the use of "relatively better performing" and "available in near-real-time" independent rainfall observations to correct real-time dynamical GFS model forecasts. In this study, we conducted additional assessment to evaluate the performance of the near-real-time product, IMERG Early. Comparison of the performance of IMERG Early with the performance of GFS would indicate to what extent the IMER Early products could be used for

calibration of GFS forecasts. We have added a new paragraph in the Conclusions section (paragraph 3) as well as additional texts in Section 2.3 to clarify this.

- The quoted statement, "The use of IMERG Early to calibrate GFS would improve GFS in terms of correlation and variability, but not in terms of bias", was removed as it was not conveying adequate information.
- The entire Conclusions Section has been edited to improve clarity.

In section 3.4, what's the forecast uncertainty? How to evaluate or quantify the uncertainty? I think the "uncertainty" in section 3.4 is only the different performance, but not uncertainty. The authors stated that the GFS forecasts show large underestimation bias for heavy rain rates. I suggest to add some explanations for the poor performance, by evaluating other variables related to the physical mechanism that affect the precipitation over the study region, or citing some relevant references.

**Response**:

- We accept the correction. We have revised it to read "Dependence of Forecast Performance on Precipitation Rate".
- There are very few studies on evaluation of GFS forecasts (see our Introduction section). We found one paper that examined the bias of GFS at high rates. The paper reported the difficulty of capturing high rain rates in GFS models. We have added this reference (see Section 3.6). Evaluation of the different error sources of GFS forecasts is outside the scope of this study, as our approach focuses on evaluation of total GFS performance (lumping together all error sources) due to limitation in our ground reference data.

In section 3.5, the authors states that the climatological bias correction approach is not effective in removing the bias in IMERG Early estimates. Why do you present the results? It is not meaningful for this manuscript. I think you could do some analysis for the effective method of bias correction to improve the IMERG Early estimates and thus to improve the GFS performance by calibration.

**Response**: We agree that it is not meaningful to present results that do not improve performance. Hence, we have removed the climatological bias correction from our evaluation. In the Conclusions section (see last paragraph), we have recommended the need for identifying appropriate bias calibration techniques, and suggested some potential methodologies that could be explored.

In addition, the introduction should be improved seriously. For example, the current studies on the evaluation of GFS forecasts and its performance on global scale or other regions should be added.

**Response**: We have added literature review of GFS performance evaluation in other regions of the world (see Paragraph 4 of the Introduction Section).

Minor comments:
The abstract should be carefully revised. For example, it should not include the detail introduction of study basin.

**Response:** We have revised the abstract, and also removed the details in the introduction of the study basin.

The resolution of GFS forecasts and Satellite Precipitation Products are not consistent, how do you deal with them? The authors do not describe any information about this.

**Response**: We have added statements describing the methodology used to bring both products to the same resolution (see paragraph 1 of Section 2.5).

Line 76: remove ")"

**Response**: Done. Thank you.

Figure 1: I suggest to add legend for the drainage basin, or use the appropriate color for the boundary of the sub-basin

**Response:** Done. Thank you.

What is R in Figure 2? Please classify.

**Response:** It was correlation, however, we have removed it now as it is not that meaningful.

Line 235: "How well is the annual precipitation total forecasted in each dam watershed?" I do not understand, please classify.

**Response:** Here, we aggregate the 1-day lead GFS forecasts to annual time scale and compared the results against corresponding annual precipitation estimates from IMERG Final. To improve clarity of the text, we have revised the section heading as "How well do GFS forecasts capture annual rainfall?" and also added a clarifying text (see first sentence of that section).

---

## Author Response (AR2)

**Response to Reviewer's Comments**

**Comment**: The authors have successfully addressed each reviewer's comments, which has significantly improved the manuscript. However, I do have some concerns, which I think authors can address before the manuscript is accepted.

**Response**: Thank you for your additional comments. Please find below our response summarizing how we addressed these comments.

**Comment**: In the abstract and Lines 442-445, authors recommend that post-processing calibration techniques via the use of near-real time products, (e.g., IMERG Early) that could improve the performance of GFS. However, I still think that the authors could not simply state that the performance of post-processed GFS could be improved over some regions via comparison of the performance of GFS and IMERG Early productions, despite the performance of IMERG Early productions is better than that of GFS. Please clarify.

**Response**: Agreed. We have removed the statement from the abstract.

**Comment**: Figure 8, please add the significance level for the R.

**Response:** For Figure 8, we have added the standard error. The p-values in all cases are less than 0.001; we prefer to show the standard error as these may be more meaningful.

**Comment**: Lines 445-448, authors suggested some post-processing calibration techniques, are they performed well over the focused region or other regions? Or are they widely used? Please add some references for them.

**Response**: We have modified one of the statements to indicate that we are recommending **testing** the suitability of IMERG Early for use in post-processing. We have also added the quested references. Some of these methods are basic (e.g., multiplicative bias correction) but some are cutting-edge (e.g, AI-based methods).